# Low Dietary n-6/n-3 PUFA Ratio Regulates Meat Quality, Reduces Triglyceride Content, and Improves Fatty Acid Composition of Meat in Heigai Pigs

**DOI:** 10.3390/ani10091543

**Published:** 2020-09-01

**Authors:** Qiuyun Nong, Liyi Wang, Yanbing Zhou, Ye Sun, Wentao Chen, Jintang Xie, Xiaodong Zhu, Tizhong Shan

**Affiliations:** 1College of Animal Sciences, Zhejiang University, Hangzhou 310058, China; qynong@zju.edu.cn (Q.N.); 21917014@zju.edu.cn (L.W.); 11817022@zju.edu.cn (Y.Z.); 11817020@zju.edu.cn (Y.S.); chen.wentao@zju.edu.cn (W.C.); 2The Key Laboratory of Molecular Animal Nutrition, Ministry of Education, Hangzhou 310058, China; 3Zhejiang Provincial Laboratory of Feed and Animal Nutrition, Hangzhou 310058, China; 4Shandong Chunteng Food Co. Ltd., Zaozhuang 277500, China; sdctxjt@163.com (J.X.); sdtzZxd009@163.com (X.Z.)

**Keywords:** meat quality, Heigai pig, polyunsaturated fatty acids, growth performance

## Abstract

**Simple Summary:**

The content of n-3 and n-6 polyunsaturated fatty acids (PUFA) in pork is linked to human health. Dietary fatty acid composition especially the n-6/n-3 PUFA ratio can affect meat quality in lean breeds of pigs. However, the effects of different dietary fatty acid composition on Chinese indigenous pig breeds are still poorly understood. In the current study, Heigai pigs were fed with different n-3/n-6 PUFA ratios diets (8:1, 5:1, and 3:1) to investigate the effects of dietary supplementation with different n-6/n-3 PUFA ratios on growth performance, meat quality, and fatty acid profiles. Low dietary n-6/n-3 PUFA ratio regulated pH and meat color of longissimus dorsi muscle (LDM), reduced triglyceride and total cholesterol contents, and enhanced the deposition of n-3 PUFA of LDM and subcutaneous adipose tissue (SAT). This study can provide an experimental reference for producing functional pork (e.g., better fatty acid composition) with the advantages of Chinese local breed pigs.

**Abstract:**

The objective of this study was to investigate the effects of dietary supplementation with different n-6/n-3 polyunsaturated fatty acid (PUFA) ratios on growth performance, meat quality, and fatty acid profile in Heigai pigs. A total of 54 Heigai finishing pigs (body weight: 71.59 ± 2.16 kg) were randomly divided into three treatments with six replications (three pigs per replication) and fed diets containing different n-6/n-3 PUFA ratios: 8:1, 5:1, and 3:1. Pigs fed the dietary n-6/n-3 PUFA ratio of 8:1 had the highest feed to gain ratio (*p* < 0.01), carcass weight (*p* < 0.05), redness a* (*p* < 0.01), and yellowness b* (*p* < 0.01). Fatty acid compositions in longissimus dorsi muscle (LDM) and subcutaneous adipose tissue (SAT) were significantly changed (*p* < 0.01). Notably, the meat from the pigs fed with the low dietary n-6/n-3 PUFA ratio had higher n-3 PUFA contents (*p* < 0.01) and lower n-6/n-3 PUFA ratio (*p* < 0.01). The triglyceride and total cholesterol contents were significantly decreased in SAT from the pigs fed with dietary n-6/n-3 PUFA ratios of 5:1 (*p* < 0.05) and 3:1 (*p* < 0.01). Reducing n-6/n-3 PUFA ratio upregulated the expression of HSL (*p* < 0.05), CPT1 (*p* < 0.01), and FABP4 (*p* < 0.01) but downregulated ATGL (*p* < 0.01) expression. These results demonstrate that the lower n-6/n-3 PUFA ratio regulates meat quality and enhances the deposition of n-3 PUFA in Heigai pigs.

## 1. Introduction

Pork is one of the most consumed meats worldwide and is an important and widely available protein component of human diet due to its unique chemical composition, nutritional value, and the content of balanced protein [1]. The quality and safety of pork are closely related to human health [2,3]. Hence, it is of great significance to improve pork quality and provide healthy and safe pork for humans.

Pork quality is affected by multiple interactive factors, including genetics, nutrition, management practices, slaughtering procedures, and handling of the porcine carcass [1,4]. Importantly, previous studies have demonstrated that dietary fatty acid composition plays a vital role in regulating the nutritional quality of pork [5,6]. Fatty acids, especially essential fatty acids, are basic and important nutrients in the human diet. The content of n-3 and n-6 polyunsaturated fatty acids (PUFA) and their proportions are linked to human diseases, including obesity, cardiovascular, cerebrovascular diseases, depression, and cancer, and high n-3 PUFA content and low n-6/n-3 PUFA ratio are more beneficial to human health [7,8]. Notably, pork has a high n-6/n-3 PUFA ratio due to typical feeding practices [9]. The content of n-3 PUFA in pork can be regulated by genetic modification and nutritional regulation, which can make pork more in line with human dietary demand for n-3 PUFA [5,10]. Nutritional regulation is a safe and acceptable strategy to improve fatty acid composition and is also a hot spot of research at this stage. Recent studies have found that fatty acids especially n-6/n-3 PUFA ratio could influence the fatty acid composition and meat quality in lean breeds of pigs [5,6,11]. Breed is also an important factor that can significantly affect the fatty acid composition of pork. Compared with the typical lean pig breeds, some Chinese indigenous pig breeds have certain advantages in fatty acid composition, better flavor, and higher PUFA [12]. Heigai pig, one of the Chinese local fatty breeds, exists in Shandong province and has the specific characteristics of prolificacy, crude feed tolerance, tender meat, strong disease resistance, strong adaptability to the environment, and so on, and is worth studying. However, research on indigenous pig breeds are few and the effects of dietary fatty acid composition on indigenous pig breeds are still poorly understood. Furthermore, no studies have been done to examine the regulation of dietary fatty acid composition on growth performance, meat quality, and fatty acid profiles in Heigai pigs.

In this study, we investigated the effects of dietary supplementation with different n-6/n-3 PUFA ratios on growth performance, meat quality, fatty acid composition, and lipid metabolism-related gene expression in Heigai pigs. Our results can provide an experimental reference for producing functional pork (e.g., better fatty acid composition) with the advantages of Chinese local breed pigs.

## 2. Materials and Methods

### 2.1. Experimental Design and Diets

All procedures and housing were approved by the Zhejiang University Animal Care and Use Committee. The ethical committee number for the study is ZJU20170466. A total of 54 Heigai finishing pigs (body weight: 71.59 ± 2.16 kg) were randomly divided into 3 treatments with 6 replications (3 pigs per replication) and were fed with different n-6/n-3 PUFA ratio diets: 8:1, 5:1, and 3:1 (the fatty acid profiles of the different diets were shown in Table A1). The composition and nutritional levels of the diet were shown in Table 1. The pigs were group-housed fed with the corresponding diet for 75 days (five days pre-feeding period and seventy days formal test period). The growth performance, including body weight (BW) and feed intake of each pen were recorded, and average daily gain (ADG), average daily feed intake (ADFI), ratio feed to gain (F/G), and feed conversion ratio of each pen, were calculated. At the end of the feeding, 1 pig (close to the average body weight) per replication were selected from each group and fasted for 12 h to then be humanely sacrificed.

### 2.2. Sample Collection

Samples of longissimus dorsi muscle (LDM) were obtained from the 3rd to 11th rib for meat quality measurement. About 150 g LDM and subcutaneous adipose tissue (SAT) at the 13th rib on right side carcass were rapidly collected, frozen in liquid nitrogen immediately, and subsequently stored at −80 °C for fatty acid composition and gene expression analysis. A block of (fat or muscle tissue) was removed from the body (not more than 0.25 cm^2^) and was put into an fixative solution (10% formalin) for hematoxylin-eosin staining.

### 2.3. Carcass Traits Measurement

Slaughter weight, carcass weight, dressing percentage, carcass length, backfat thickness, skin thickness, loin muscle area, and lean percentage were measured after slaughter. Backfat thickness measurements were taken at the midline with a sliding caliper. Mean backfat thickness was calculated by averaging the scores of three regions at the first rib, last rib, and last lumbar vertebrae of the right carcass sides.

### 2.4. Determination of Meat Quality

The pH value of each sample was measured three times between the fourth and fifth lumbar vertebrae in the left side LDM at 45 min and 24 h after slaughter using a pH-meter (PH-STAR, MATTHAUS, Germany). The drip loss was determined by suspending muscle samples standardized for surface area in plastic bags at 4 °C for 24 h. Drip loss was expressed as a percentage of the initial weight [13]. Marbling scores and meat color scores were scored at 45 min after slaughter according to reference standards (NPPC-1994, America). There are 5 scores on the NPPC colorimetric board, which are arranged from light to dark for the quantitative evaluation of meat color: 1 point = off-white (abnormal meat color); 2 points = light gray (prone to abnormal meat color); 3 points = normal bright red; 4 points = slightly dark red (normal meat color); 5 points = dark purple (abnormal meat color). There are 5 scores on the NPPC marbling measurement board: 1 point = little fat trace; 2 points = fat trace; 3 points = fat; 4 points = a lot of fat; 5 points = excess fat. Meat color was respectively measured 45 min and 24 h after slaughter at the surface of a 2 -cm-thick boneless loin chop using a Minolta CM-2002 (Osaka, Japan) spectrophotometer with CIE lab color system: L* (lightness), a* (red-green) and b* (yellow-blue).

### 2.5. Hematoxylin-Eosin Staining

The tissues were taken from the fixative solution for paraffin embedding and microtome sectioning. The paraffin sections were stained by hematoxylin-eosin staining kit (Servicebio, Wuhan, China). Briefly, after deparaffinization and rehydration, sections were stained with hematoxylin solution for 5 min followed by being immersed in 1% acid ethanol (1% HCl in 70% ethanol) for 10 s and then rinsed in water for 1 h. Finally, the sections were stained with eosin solution for 3 min and followed by dehydration with graded alcohol and clearing in xylene [14]. The slides were examined and photographed using an Olympus BX61 fluorescence microscope (Japan). Observed morphological structure of cells. The size of cells (4 random fields of equal area) was analyzed by Image-Pro Plus 6.0 software.

### 2.6. Determination of Triglyceride, Total Cholesterol, and Non-Esterified Fatty Acid

The contents of triglyceride (TG), total cholesterol (TC), and non-esterified fatty acid (NEFA) in LDM and SAT were determined by commercial kits (TG, A110-1-1; TC, A111-1-1; NEFA, A042-2-1) bought from Nanjing Jiancheng Institute of Bioengineering.

### 2.7. Fatty Acid Composition Analysis

Fatty acid profiles of LDM and SAT samples were determined by gas chromatography as previously reported [15]. Briefly, samples were extracted with a mixture of chloroform and methanol (2:1; *v*/*v*) according to the methods previously described [16]. Then, total fat was converted into fatty acid methyl esters (FAMEs) and determined by gas chromatography, which was equipped with a capillary column. The GC Chem Station software was used to separate FAMEs. The FAMEs profiles of the samples were compared with FAMEs standards to identify the fatty acids in SAT and LDM. Fatty acid content was expressed as a percentage of total fatty acids.

### 2.8. Total RNA Extraction, Reverse Transcription, and Quantitative Real-Time PCR

Total RNA was prepared from the LDM and SAT using TRIzol reagent (Thermo Fisher, Waltham, MA, USA) according to the instruction of the manufacture as previously described [17]. The concentration and purity of the total RNA were measured using the NanoDrop 2000 instrument (Gene Company Limited, Hong Kong, China). About 2 μg of total RNA samples were subjected to reverse transcription with the use of random primers and a ReverAid First Strand cDNA Synthesis Kit (Thermo Fisher). The qPCR was performed by the BioRad CFX96Touch Fast Real-Time PCR System and FastStart Universal SYBR Green Master (ROX, Shanghai, China). The primers used for qPCR were: Glyceraldehyde-3-phosphate dehydrogenase (GPDH; GPDH-F: AAGGAGTAAGAGCCCCTGGA; GPDH-R: TCTGGGATGGAAACTGGAA); adipose triglyceride lipase (ATGL; ATGL-F: TCACCAACACCAGCATCCA; ATGL-R: GCACATCTCTCGAAGCACCA); Hormone-sensitive lipase (HSL-F: GCAGCATCTTCTTCCGCACA; HSL-R: AGCCCTTGCGTAGAGTGACA); Carnitine palmitoyltransferase 1 (CPT1-F: ATGGTGGGCGACTAACT; CPT1-R: TGCCTGCTGTCTGTGAG); Fatty acid binding protein 4 (FABP4-F: GGGACATCAAGGAGAAGC; FABP4-R: ACCGTGTTGGCGTAGAG). The relative changes in gene expression normalized against GAPDH rRNA as the internal control (analyzed by the 2−ΔΔCT method). Dietary n-6/n-3 PUFA ratio of 8:1 was used as the control diet to calculate relative gene expression.

### 2.9. Statistical Analysis

Data were analyzed by one-way ANOVA of SPSS 20.0 (IBM-SPSS Inc., Chicago, IL, USA) and compared using Duncan posthoc test. *p* < 0.05 was considered to be statistically significant.

## 3. Results

### 3.1. Effects of Dietary n-6/n-3 PUFA Ratio on the Growth Performance of Heigai Pigs

The results showed that the total weight gain, ADG, and ADFI have no significant difference between treatments. The lower the dietary n-6/n-3 PUFA ratio considered, the higher the value of F:G ratio was observed (*p* < 0.01) (Table 2).

### 3.2. Effects of Dietary n-6/n-3 PUFA Ratio on the Carcass Quality of Heigai Pigs

Pigs fed the dietary n-6/n-3 PUFA ratio of 8:1 had higher carcass weight than pigs fed the dietary n-6/n-3 PUFA ratio of 5:1. However, carcass traits including slaughter weight, dressing percentage, carcass length, backfat thickness, skin thickness, loin muscle area, and lean percentage were not significantly affected by the treatments (*p* > 0.05) (Table 3).

### 3.3. Comparisons of Meat Quality among Different Dietary n-6/n-3 PUFA Ratio Groups

We determined meat quality traits in Heigai pigs fed with different n-6/n-3 PUFA ratio diets (Table 4). Pigs fed the dietary n-6/n-3 PUFA ratio of 8:1 had the lowest pH (24 h) (*p* < 0.05) and L* (24 h) (*p* < 0.05), but had the highest ΔpH (*p* < 0.01), a* (45 min) (*p* < 0.01), a* (24 h) (*p* < 0.01), b* (24 h) (*p* < 0.01), Δa* (*p* < 0.01), and Δb* (*p* < 0.01). The remainder of the parameters evaluated (pH (45 min), drip loss, marbling score, meat color score, intramuscular fat, 45 min L*, 45 min b*) were not significantly affected by the treatments (*p* > 0.05).

### 3.4. Effects of Dietary n-6/n-3 PUFA Ratio on Morphological Structure and the Contents of TG, TC and NEFA in LDM and SAT

We found no significant differences in morphological structure and average cell size among the three treatments in LDM and SAT (Figure 1a). In LDM and SAT, TG contents were decreased significantly in the pigs fed the dietary n-6/n-3 PUFA ratio of 5:1 (*p* < 0.05) and 3:1 (*p* < 0.01) (Figure 1b). The highest TC content in SAT (*p* < 0.01) has been found in the pigs fed the dietary n-6/n-3 PUFA ratio of 8:1. However, the TC content of LDM has no significant difference (Figure 1c). Besides, NEFA was not affected by the dietary n-6/n-3 PUFA ratio in both LDM and SAT (Figure 1d) (*p* > 0.05).

### 3.5. Effects of Dietary n-6/n-3 PUFA Ratio on the Fatty Acid Profiles in LDM

To explore the changes of the overall fatty acid profiles in LDM from pigs fed with different dietary n-6/n-3 PUFA ratios, we analyzed the composition of fatty acids (Table 5). Pigs fed the dietary n-6/n-3 PUFA ratio of 8:1 has the highest contents in C15:0 (*p* < 0.05), C16:0 (*p* < 0.05), C18:2 (n-6 PUFA; *p* < 0.05), C20:1 (n-9 PUFA; *p* < 0.01), C22:0 (*p* < 0.05), C22:1 (n-9 PUFA; *p* < 0.05), n-6 PUFA (*p* < 0.05), but showed the lowest contents in C16:1 (*p* < 0.05), C18:0 (*p* < 0.05), C18:3 (n-3 PUFA; *p* < 0.01), C20:3 (n-3 PUFA; *p* < 0.01), and n-3 PUFA (*p* < 0.01). The highest contents of C16:1 (*p* < 0.05), C18:0 (*p* < 0.05), C18:3 (n-3 PUFA; *p* < 0.01), C20:3 (n-3 PUFA; *p* < 0.01), PUFA (*p* < 0.05), and n-3 PUFA (*p* < 0.01), but the lowest contents of C15:0 (*p* < 0.05), C16:0 (*p* < 0.05), C20:1 (n-9 PUFA; *p* < 0.01), C22:0 (*p* < 0.05), and C22:1 (n-9 PUFA; *p* < 0.05) have been found in the pigs fed the dietary n-6/n-3 PUFA ratio of 3:1. Pigs fed the dietary n-6/n-3 PUFA ratio of 5:1 had the lowest contents in C18:2 (n-6 PUFA; *p* < 0.01), PUFA (*p* < 0.05), and n-6 PUFA (*p* < 0.05). The ratio of n-6/n-3 PUFA decreased in turn from the pigs fed the dietary n-6/n-3 PUFA ratio of 8:1 to 3:1. Pigs fed the diet with the 3:1 ratio showed the lowest n-6/n-3 ratios (*p* < 0.01) followed by the 5:1 ratio (*p* < 0.01), and the 8:1 ratio group has the highest n-6/n-3 ratios (*p* < 0.01). The remainder of the fatty acid contents was not significantly affected by the treatments (*p* > 0.05).

### 3.6. Dietary n-6/n-3 PUFA Ratio Alters the Fatty Acid Profiles in SAT

Similar to fatty acid profiles of the LDM, UFA was the most abundant. C18:3 (n-3 PUFA, *p* < 0.01), C20:3 (n-3 PUFA, *p* < 0.01), and n-3 PUFA were increased in the pigs fed the low dietary n-6/n-3 PUFA ratio. C18:3 (n-3 PUFA, *p* < 0.01), C20:3 (n-3 PUFA, *p* < 0.01), and n-3 PUFA (*p* < 0.01) in the pigs fed the dietary n-6/n-3 PUFA ratio of 3:1 was the highest. However, no significant differences were found in n-6 PUFA contents among the three treatments. The ratio of n-6/n-3 PUFA also decreased in turn from the pigs fed the dietary n-6/n-3 PUFA ratio of 8:1 to 3:1 in SAT. Additionally, the pigs fed the dietary n-6/n-3 PUFA ratio of 5:1 (*p* < 0.05) and 3:1 (*p* < 0.01) showed lower n-6/n-3 PUFA ratio in SAT. However, different from LDM, the n-6/n-3 PUFA ratio in SAT has no significant difference between the pigs fed the dietary n-6/n-3 PUFA ratio of 5:1 and 3:1 (*p* > 0.05), and the remainder of the fatty acid contents were not significantly affected by the treatments (*p* > 0.05) (Table 6).

### 3.7. Effects of Dietary n-6/n-3 PUFA Ratio on Expression of Lipid Metabolism Related Genes in LDM and SAT

We analyzed the changes in the expression of lipid metabolism-related genes, including ATGL, HSL, FABP4, and CPT1 (Figure 2). ATGL was expressed at a higher level in the pigs fed the dietary n-6/n-3 PUFA ratio of 8:1 than 5:1 (*p* < 0.01) and 3:1 (*p* < 0.01) in SAT but has no significant difference in LDM (Figure 2a). Compared with the pigs fed the dietary n-6/n-3 PUFA ratio of 8:1, ratio of 5:1 (*p* < 0.05), and 3:1 (*p* < 0.05) groups have an upregulated expression of HSL in LDM (Figure 2b). However, HSL expression in SAT was not affected (Figure 2b). In SAT, the pigs fed the dietary n-6/n-3 PUFA ratio of 3:1 has higher CPT1 expression than the dietary n-6/n-3 PUFA ratio of 5:1 (*p* < 0.01) and 8:1 (*p* < 0.05) (Figure 2c). In LDM, pigs fed the dietary n-6/n-3 PUFA ratio of 5:1 showed the lowest CPT1 expression (*p* < 0.05) (Figure 2c). The pigs fed the dietary n-6/n-3 PUFA ratio of 5:1 (*p* < 0.01) and 3:1 (*p* < 0.05) showed high FABP4 expression in SAT but has no significant difference in LDM (Figure 2d).

## 4. Discussion

In this study, we investigated the effects of dietary supplementation with different n-6/n-3 PUFA ratios on the growth performance, meat quality, and fatty acid profiles in Heigai pigs. We found that dietary supplementation with a lower n-6/n-3 PUFA ratio increased the ratio of feed to gain. Notably, dietary n-6/n-3 PUFA ratio regulated the pH and meat color measured by CIE lab color system. However, sensory evaluation of the meat color was not affected. Moreover, dietary n-6/n-3 PUFA ratio affected the fatty acid profiles in LDM and SAT of Heigai pigs. The contents of the TG and TC as well as the expression of lipid metabolism-related genes in LDM and SAT were regulated by dietary n-6/n-3 PUFA ratio.

Dietary supplementation with lower n-6/n-3 PUFA ratio increased the ratio of feed to gain and decreased the carcass weight. However, a previous study showed that a diet with a lower n-6/n-3 PUFA ratio, rich in n-3 PUFA, was beneficial for the growth performance and health of animals [18]. Although the difference was not significant, the backfat thickness was reduced in our results. These results indicate that the dietary n-6/n-3 PUFA ratio can affect the increase of carcass weight by reducing the backfat contents, causing the increased F: G ratio with a lower n-6/n-3 PUFA ratio.

In this study, pigs fed the dietary n-6/n-3 PUFA ratio of 3:1 reduced the ΔpH, a* (45 min), a* (24 h), b* (24 h), Δa*, and Δb*, but increased the pH (24 h), and L* (24 h). The results of ΔpH, Δa*, and Δb* suggest that lower dietary n-6/n-3 PUFA ratio protects against the rate of pH drop, and the rate of a* and b* raise. Likewise, a previous study reported that there was a reduction in b* after feeding a high n-3 PUFA content diet [19]. However, other studies showed that a higher n-3 PUFA diet had no significant effect on pH, L*, a*, and b* [20,21,22]. These differences might be due to the different breeds and diets used in the current and previous studies. These results suggested that supplementation with high n-3 PUFA and low n-6/n-3 PUFA ratio may regulate the pH and meat color of LDM. Moreover, we found no significant differences in sensory evaluation of meat color, which prompted us to assume that consumers may not feel the change in meat color.

We also found that high n-3 PUFA and low n-6/n-3 PUFA ratio diet reduced TC and TG contents. These results were consistent with the previous findings that n-3 PUFA could help to reduce TG and TC contents [23,24]. These data suggest that high n-3 PUFA and low n-6/n-3 PUFA ratio diets could help us to produce pork meat with low TC and TG contents.

Our results also demonstrate that the dietary n-6/n-3 PUFA ratio altered the fatty acid profiles in SAT and LDM of Heigai pigs, which are consistent with the results of previous studies that dietary fatty acid composition can affect serum, fat, muscle, and liver fatty acid composition [5,25,26]. Consistent with the previous studies [27,28,29] that a large alternation in C18: 3 (n-3 PUFA) caused the main changes in the total amount of n-3 PUFA in the current study. In addition, C16: 0 and C18: 1 were two of the most abundant, and UFA was more than SFA in LDM and SAT. These results are in agreement with the results reported by *A. de Tonnac* [5]. Both in LDM and SAT, the levels of SFA, UFA, and MUFA were not affected. Consistent with our results, a previous study also showed that SFA, UFA, and MUFA in LDM and SAT were not affected by dietary n-3 PUFA contents [5]. However, *Monique J. Van Oeckel* had reported the inconsistent result that SFA, UFA, and MUFA had significant differences after feeding with a high n-3 PUFA content diet [30]. Moreover, the n-6/n-3 PUFA ratio in LDM and SAT were positively correlated with the ratio in diets. Consistent with previous studies [6,31], the n-6/n-3 PUFA ratio in LDM was higher than that in SAT, suggesting that the fatty acid metabolism in LDM and SAT are different. In addition, fatty acid absorption and deposition are different in different breeds and tissues. Not only n-6 PUFA and n-3 PUFA contents, but also the contents of the other fatty acids including C15:0, C16:0, C16:1, C18:0, C20:1 (n-9 PUFA), C22:0, and C22:1 (n-9 PUFA) were affected by dietary n-6/n-3 PUFA ratio, suggesting that the metabolism of fatty acids may be affected by dietary n-6/n-3 PUFA ratio.

Fatty acids are released from intracellular fat stores by the action (or signaling) of ATGL [32]. Our results indicate that a lower dietary n-6/n-3 PUFA ratio may reduce fatty acids release by reducing ATGL expression in SAT. HSL, an intracellular enzyme, regulates the release of NEFA from lipid stores [33]. We found that low n-6/n-3 PUFA ratio diet upregulated HSL expression in LDM. Previous studies had also reported that lower dietary n-6/n-3 PUFA ratio increased HSL expression level in SAT [6]. Low n-6/n-3 PUFA ratio diet also enhances FABP4 expression in SAT, which against the previous study that different n-6/n-3 PUFA diets, did not affect the expression of FABP4 [34]. The expression of CPT1 was increased in the Heigai pigs fed the dietary n-6/n-3 PUFA ratio of 3:1 in SAT and LDM. This result was consistent with the previous studies that n-3 PUFA enhanced the expression of CPT1 [24]. ATGL, HSL, CPT1, and FABP4 expression are related to lipid metabolism especially lipolysis. Hence, a lower dietary n-6/n-3 PUFA ratio may enhance lipolysis and affect fatty acid metabolism through increasing the expression level of HSL, FABP4, and CPT1. The enhanced fat catabolism through increasing expression level of HSL, FABP4, and CPT1 may be the reason why lower dietary n-6/n-3 PUFA ratio groups showed lower F: G ratio, carcass weight, and TG content. However, the mechanism is not yet clear and needs further investigation.

## 5. Conclusions

In conclusion, we demonstrate that dietary fatty acid composition used in this study regulated the pH and meat color of LDM and improved fatty acid composition in meat. Moreover, the low dietary n-6/n-3 PUFA ratio improved the fatty acid profiles of pork. Our results suggest that combining the dietary fatty acid composition with the advantages of Chinese local breeds can improve the fatty acid composition of pork, make pork a better source of fatty acids (high n-3 PUFA content and low n-6/n-3 PUFA ratio), and improve the meat quality. Further studies should investigate the potential molecular mechanism and provide more in-depth practical knowledge on functional pork production.

## Figures and Tables

**Figure 1 animals-10-01543-f001:**
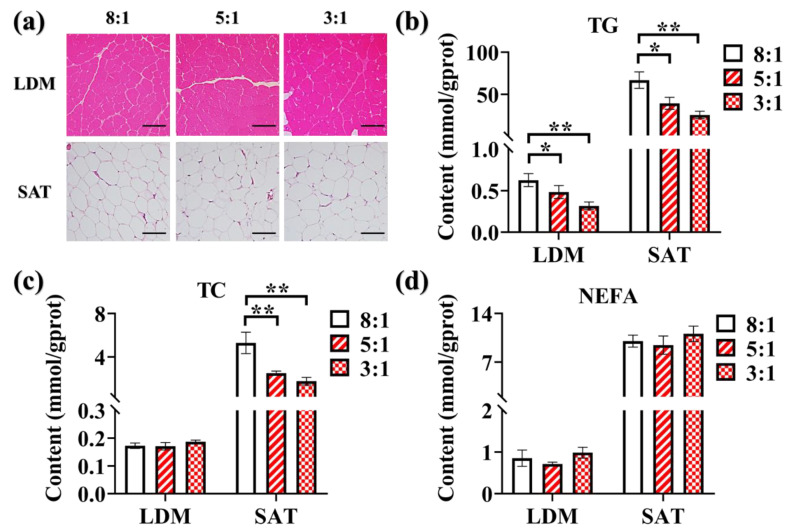
Morphological structure and the contents of triglyceride (TG), total cholesterol (TC), and non-esterified fatty acid (NEFA) in Longissimus dorsi muscle (LDM) and subcutaneous adipose tissue (SAT) from Heigai pigs fed with different diets. (**a**) Histological images (×100) of Hematoxylin-Eosin staining showed the influences of different n-6/n-3 PUFA ratio diet on LDM and SAT morphological structure of Heigai pigs; scale bar = 200μm. The influences of different n-6/n-3 PUFA ratio on TG (**b**), TC (**c**), and NEFA (**d**) contents in LDM and SAT of Heigai pigs. ‘8:1, 5:1, 3:1’: different n-6/n-3 PUFA ratio diets. Results were presented by mean ± standard error of the mean (SEM) (n = 6). ‘*’ *p* < 0.05; ‘**’ *p* < 0.01.

**Figure 2 animals-10-01543-f002:**
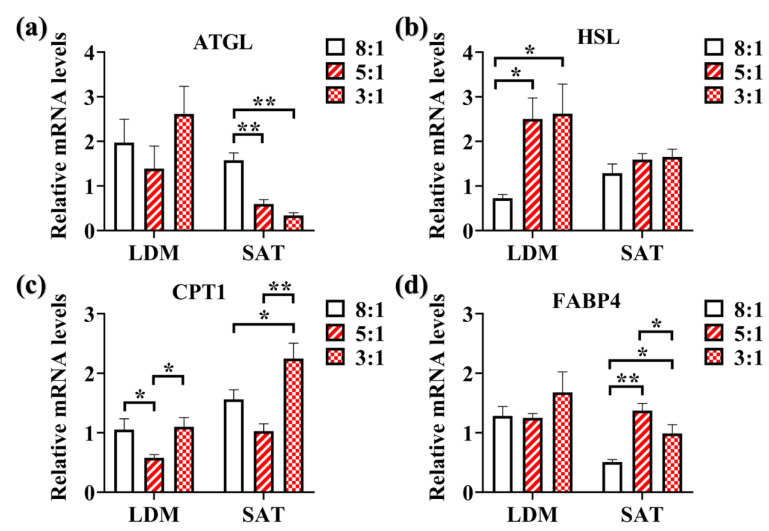
The expression of relative genes in longissimus dorsi muscle (LDM) and subcutaneous adipose tissue (SAT) from Heigai pigs fed with different diets. The influences of different n-6/n-3 polyunsaturated fatty acids (PUFA) ratio on adipose triglyceride lipase (ATGL) (**a**), hormone-sensitive lipase (HSL) (**b**), carnitine palmitoyltransferase 1 (CPT1) (**c**), and fatty acid binding protein 4 (FABP4) (**d**) expression in LDM and SAT of Heigai pig. ‘8:1, 5:1, 3:1’: different n-6/n-3 PUFA ratio diets. Results were presented by mean ± standard error of the mean (SEM) (*n* = 4, number of replicates). ‘*’ *p* < 0.05; ‘**’ *p* < 0.01.4.

**Table 1 animals-10-01543-t001:** Ingredient composition and nutritional levels of the diet (air-dry basis).

Ingredient	Content (%)			Nutritional Levels ^2^	Content (%)
	8:1	5:1	3:1 ^1^		
Corn	50.00	50.00	50.00	Crude protein	11.85 ± 0.76
Soybean meal	4.00	4.00	4.00	Crude fiber	12.20 ± 0.61
Wheat bran	17.00	17.00	17.00	Crude fat	5.63 ± 0.15
Calcium hydrogen phosphate	0.50	0.50	0.50		
Limestone powder	1.00	1.00	1.00		
Dried ground hay	23.70	23.70	23.70		
Soybean oil	2.30	1.50	0.15		
Linseed oil	0.70	1.50	2.85		
Salt	0.30	0.30	0.30		
Premix ^1^	0.50	0.50	0.50		
Digestive energy, MJ/kg	12.2296	12.2299	12.2347		

Each kilogram of the premix contains 800–1.6 million IU vitamin A acetate, 3.00 mg cyanocobalamin, 650–125 million IU vitamin D3, 5.50 × 103 mg nicotinamide, 9 × 103 mg α-tocopheryl acetate, 4.00 × 103 mg D-calcium pantothenate, 350 mg menadione, 17.5 mg D-biotin, 1.50 × 103 mg riboflavin, 0.18 × 103–7.00 × 103 mg copper, 0.75 × 104–2.00 × 104 mg iron, 3.75 × 103–7.5 × 103 mg manganese, 1.00 × 104–4.00 × 104 mg zinc, 50–100 mg selenium, 15.0–30.0 mg iodine, etc. The rest is limestone powder. ^1^ ‘8:1, 5:1, 3:1’: different n-6/n-3 fatty acid ratios diets. ^2^ Results of nutritional levels were presented by mean ± standard deviation (SD) (*n* = 3, number of replicates).

**Table 2 animals-10-01543-t002:** Effects of dietary n-6/n-3 polyunsaturated fatty acids (PUFA) ratio on the growth performance of Heigai pigs.

Growth Performance	Values of Each Group			SEM	*p*-Value
	8:1	5:1	3:1 ^1^		
Total weight gain (kg)	31.53 ± 7.69	28.25 ± 8.47	28.72 ± 7.36	1.07	0.406
ADG (kg/d)	0.42 ± 0.10	0.38 ± 0.11	0.38 ± 0.10	0.01	0.400
ADFI (kg/d)	2.02 ± 0.49	2.00 ± 0.60	2.07 ± 0.53	0.07	0.936
F:G ratio	4.80 ± 0.01 ^C^	5.27 ± 0.19 ^B^	5.40 ± 0.01 ^A^	0.18	<0.001

Results were presented by mean ± standard deviation (SD). SEM = standard error of the mean; (*n* = 18, number of replicates); ADG = average daily gain; ADFI = average daily feed intake; F:G ratio = ratio of feed to gain. ^A,B,C^ Different superscripts within a row indicate significant differences (*p* < 0.01). ^1^ ‘8:1, 5:1, 3:1’: different n-6/n-3 PUFA ratio diets.

**Table 3 animals-10-01543-t003:** Effects of dietary n-6/n-3 PUFA ratio on the carcass traits of Heigai pigs.

Carcass Quality Traits	Values of Each Group			SEM	*p*-Value
	8:1	5:1	3:1 ^1^		
Slaughter weight (kg)	102.50 ± 5.25	94.58 ± 8.78	100.75 ± 10.49	2.40	0.266
Carcass weight (kg)	79.97 ± 4.01 ^a^	72.72 ± 6.13 ^b^	76.87 ± 5.35 ^ab^	2.10	0.086
Dressing percentage (%)	78.13 ± 4.43	77.38 ± 9.49	76.55 ± 3.95	1.52	0.915
Carcass length (cm)	89.33 ± 2.66	87.17 ± 5.64	90.50 ± 5.39	1.12	0.486
Backfat thickness (mm)	45.92 ± 1.89	44.64 ± 3.76	42.85 ± 6.55	1.06	0.509
Skin thickness (mm)	5.53 ± 0.90	5.18 ± 0.99	5.10 ± 0.84	0.22	0.691
Loin muscle area (cm^2^)	24.84 ± 0.95	24.47 ± 1.54	24.97 ± 2.31	0.40	0.871
Lean percentage (%)	31.50 ± 2.16	28.45 ± 4.14	31.23 ± 3.05	0.98	0.225

Results were presented by mean ± standard deviation (SD). SEM = standard error of the mean. (*n* = 6, number of replicates). Dressing percentage (%) = Carcass weight/Slaughter weight × 100%. Lean percentage (%) = Lean weight/Carcass weight × 100%. ^a,b^ Different superscripts within a row indicate significant differences (*p* < 0.05). ^1^ ‘8:1, 5:1, 3:1’: different n-6/n-3 PUFA ratio diets.

**Table 4 animals-10-01543-t004:** Effects of dietary n-6/n-3 PUFA ratio on the meat quality of Heigai pigs

Meat Quality Traits	Values of Each Group			SEM	*p*-Value
	8:1	5:1	3:1 ^2^		
pH (45 min)	6.25 ± 0.44	6.05 ± 0.23	6.16 ± 0.31	0.08	0.619
pH (24 h)	5.58 ± 0.06 ^b^	5.70 ± 0.12 ^a^	5.71 ± 0.08 ^a^	0.04	0.047
ΔpH ^1^	0.90 ± 0.18 ^Ac^	0.31 ± 0.36 ^B^	0.43 ± 0.35 ^ABd^	0.18	0.013
Drip loss (%)	1.38 ± 0.35	1.79 ± 0.42	1.58 ± 0.72	0.12	0.410
Marbling score	3.17 ± 0.41	3.08 ± 0.38	3.00 ± 0.45	0.10	0.338
Meat color score	4.17 ± 0.75	3.83 ± 0.41	3.67 ± 0.52	0.15	0.785
Intramuscular fat (g/100 g)	4.47 ± 0.14	4.45 ± 0.10	4.48 ± 0.11	0.03	0.870
CIE					
L* (45 min)	36.77 ± 5.19	39.69 ± 4.14	40.99 ± 3.54	1.25	0.257
a* (45 min)	13.57 ± 0.93 ^A^	12.03 ± 1.55 ^AB^	10.66 ± 1.22 ^B^	0.84	0.004
b* (45 min)	8.31 ± 1.11	8.00 ± 0.79	8.33 ± 0.27	0.19	0.732
L* (24 h)	46.86 ± 1.80 ^b^	49.69 ± 3.59 ^ab^	50.74 ± 2.07 ^a^	1.16	0.054
a* (24 h)	15.84 ± 1.27 ^A^	11.12 ± 1.56 ^B^	10.32 ± 1.27 ^B^	1.72	<0.001
b* (24 h)	12.85 ± 1.35 ^A^	10.18 ± 1.61 ^B^	9.97 ± 0.56 ^B^	0.92	0.002
ΔL* ^3^	10.09 ± 3.79	10.00 ± 4.05	9.75 ± 3.57	0.90	0.987
Δa* ^4^	2.27 ± 1.59 ^A^	−0.91 ± 1.77 ^B^	−0.33 ± 1.12 ^B^	0.98	0.006
Δb* ^5^	4.54 ± 0.84 ^A^	2.19 ± 1.25 ^B^	1.65 ± 0.58 ^B^	0.89	<0.001

Results were presented by mean ± standard deviation (SD). SEM = standard error of the mean. (*n* = 6, number of replicates); ^A,B^ Different superscripts within a row indicate significant differences (*p* < 0.01). ^a,b,c,d^ Different superscripts within a row indicate significant differences (*p* < 0.05). ^1^ ΔpH = pH (45 min) − pH (24 h). ^2^ ‘8:1, 5:1, 3:1′: different n-6/n-3 PUFA ratio diets. ^3^ ΔL* = L* (24 h) − L* (45 min). ^4^ Δa* = a* (24 h) − a* (45 min). ^5^ Δb* = b* (24 h) − b* (45 min).

**Table 5 animals-10-01543-t005:** Fatty acid profiles in LDM of Heigai pigs fed with different diets.

Item	Values of Each Group (%)			SEM	*p*-Value
	8:1	5:1	3:1 ^2^		
C12:0	0.06 ± 0.02	0.05 ± 0.04	0.03 ± 0.04	0.011	0.150
C14:0	1.37 ± 0.12	1.74 ± 0.52	1.84 ± 0.47	0.144	0.140
C15:0	0.24 ± 0.21 ^a^	0.06 ± 0.13 ^b^	0.01 ± 0.01 ^b^	0.069	0.034
C16:0	42.46 ± 1.94 ^a^	38.70 ± 5.81 ^ab^	33.79 ± 6.15 ^b^	2.509	0.029
C16:1 ^1^	2.79 ± 0.86 ^b^	3.90 ± 1.14 ^a^	3.98 ± 0.41 ^a^	0.384	0.053
C17:0	0.19 ± 0.04	0.21 ± 0.07	0.25 ± 0.06	0.017	0.234
C18:0	3.64 ± 3.43 ^b^	7.61 ± 6.60 ^ab^	11.52 ± 7.37 ^a^	2.275	0.111
C18:1 (n-9)	35.69 ± 3.92	36.92 ± 3.33	35.16 ± 2.17	0.759	0.634
C18:2 (n-6)	9.52 ± 2.52 ^a^	7.04 ± 1.01 ^b^	8.63 ± 0.89 ^ab^	0.727	0.057
C18:3 (n-6)	0.04 ± 0.01	0.03 ± 0.02	0.02 ± 0.02	0.007	0.125
C18:3 (n-3)	0.91 ± 0.31 ^B^	1.11 ± 0.22 ^ABd^	1.90 ± 0.86 ^Ac^	0.301	0.016
C20:0	0.17 ± 0.09	0.16 ± 0.13	0.20 ± 0.16	0.030	0.856
C20:1 (n-9)	0.58 ± 0.14 ^A^	0.34 ± 0.27 ^AB^	0.16 ± 0.25 ^B^	0.122	0.018
C20:2 (n-6)	0.34 ± 0.11	0.29 ± 0.06	0.39 ± 0.11	0.030	0.201
C20:3 (n-6)	0.15 ± 0.06	0.18 ± 0.05	0.20 ± 0.02	0.013	0.269
C20:3 (n-3)	0.23 ± 0.21 ^B^	0.37 ± 0.20 ^ABd^	0.75 ± 0.31 ^Ac^	0.155	0.006
C20:4 (n-6)	1.12 ± 0.64	1.06 ± 0.30	1.15 ± 0.29	0.104	0.932
C20:5 (n-3)	0.09 ± 0.08	0.06 ± 0.06	0.01 ± 0.02	0.021	0.165
C22:0	0.02 ± 0.01 ^a^	0.01 ± 0.02 ^ab^	0.00 ± 0.00 ^b^	0.006	0.029
C22:1 (n-9)	0.34 ± 0.28 ^a^	0.16 ± 0.24 ^ab^	0.00 ± 0.00 ^b^	0.097	0.049
C22:6 (n-3)	0.06 ± 0.04	0.03 ± 0.03	0.03 ± 0.04	0.011	0.249
SFA	48.14 ± 2.37	48.54 ± 3.16	47.63 ± 3.47	0.716	0.874
UFA	51.86 ± 2.37	51.46 ± 3.16	52.37 ± 3.47	0.716	0.874
PUFA	12.45 ± 2.98 ^ab^	10.15 ± 1.18 ^b^	13.07 ± 1.59 ^a^	0.889	0.064
MUFA	39.41 ± 4.53	41.32 ± 3.41	39.30 ± 2.28	0.832	0.551
n-6 PUFA	11.17 ± 2.68 ^a^	8.58 ± 1.14 ^b^	10.39 ± 1.09 ^ab^	0.766	0.065
n-3 PUFA	1.28 ± 0.31 ^B^	1.57 ± 0.21 ^B^	2.69 ± 0.76 ^A^	0.428	<0.001
n-6/n-3 PUFA	8.74 ± 0.30 ^A^	5.57 ± 0.98 ^B^	4.06 ± 0.88 ^C^	1.378	<0.001

Results were presented by mean ± standard deviation (SD). SEM = standard error of the mean. (*n* = 6, number of replicates); ^a,b,c,d^ Different superscripts within a row indicate significant differences (*p* < 0.05). ^A,B^ Different superscripts within a row indicate significant differences (*p* < 0.01). ‘SFA’: saturated fatty acid, SFA = ∑ (C12:0, C14:0, C15:0, C16:0, C17:0, C18:0, C20:0, C22:0); ‘UFA’: unsaturated fatty acid, UFA =∑ (C16:1, C18:1 (n-9), C18:2 (n-6), C18:3 (n-6), C18:3 (n-3), C20:1 (n-9), C20:2 (n-6), C20:3 (n-6), C20:3 (n-3), C20:4 (n-6), C20:5 (n-3), C22:1 (n-9), C22:6 (n-3)); ‘MUFA’: monounsaturated fatty acid, MUFA = ∑ (C16:1, C18:1 (n-9), C20:1 (n-9), C22:1 (n-9)); ‘PUFA’: polyunsaturated fatty acid, PUFA = ∑ (C18:2 (n-6), C18:3 (n-6), C18:3 (n-3), C20:2 (n-6), C20:3 (n-6), C20:3 (n-3), C20:4 (n-6), C20:5 (n-3), C22:6 (n-3)). n-6 PUFA = ∑ (C18:2 (n-6), C18:3 (n-6), C20:2 (n-6), C20:3 (n-6), C20:4 (n-6)). n-3 PUFA = ∑ (C18:3 (n-3), C20:3 (n-3), C20:5 (n-3), C22:6 (n-3)). ^1^ C16:1 = ∑ (C16:1(n-7), C16:1(n-9)). ^2^ ‘8:1, 5:1, 3:1’: different n-6/n-3 PUFA ratio diets.

**Table 6 animals-10-01543-t006:** Fatty acid profiles in SAT of Heigai pigs fed with different diets.

Item	Values of Each Group (%)			SEM	*p*-Value
	8:1	5:1	3:1 ^2^		
C12:0	0.10 ± 0.01	0.11 ± 0.02	0.11 ± 0.01	0.003	0.811
C14:0	1.88 ± 0.09	1.88 ± 0.27	1.93 ± 0.17	0.045	0.881
C14:1 (n-5)	0.02 ± 0.00	0.02 ± 0.01	0.02 ± 0.00	0.001	0.561
C15:0	0.08 ± 0.03	0.06 ± 0.03	0.07 ± 0.01	0.005	0.393
C16:0	28.18 ± 1.11	30.48 ± 7.20	27.23 ± 1.35	1.008	0.420
C16:1 ^1^	2.44 ± 0.39	3.06 ± 0.49	2.62 ± 0.56	0.180	0.113
C17:0	0.39 ± 0.10	0.37 ± 0.21	0.39 ± 0.05	0.032	0.972
C18:0	15.68 ± 1.38	11.88 ± 5.57	14.82 ± 2.82	1.150	0.208
C18:1 (n-9)	33.12 ± 3.57	34.17 ± 4.35	31.33 ± 2.80	0.855	0.413
C18:2 (n-6)	13.74 ± 2.79	12.92 ± 4.17	14.83 ± 2.17	0.744	0.584
C18:3 (n-6)	0.05 ± 0.02	0.06 ± 0.03	0.07 ± 0.01	0.005	0.643
C18:3 (n-3)	2.64 ± 0.72 ^B^	3.47 ± 1.35 ^ABd^	4.72 ± 0.63 ^Ac^	0.600	0.007
C20:2(n-6)	0.86 ± 0.17	0.66 ± 0.33	0.85 ± 0.15	0.060	0.279
C20:3 (n-6)	0.11 ± 0.02	0.10 ± 0.05	0.11 ± 0.01	0.007	0.484
C20:3 (n-3)	0.33 ± 0.11 ^B^	0.43 ± 0.13 ^ABd^	0.62 ± 0.15 ^Ac^	0.090	0.005
C20:4 (n-6)	0.22 ± 0.04	0.24 ± 0.09	0.19 ± 0.03	0.020	0.328
C20:5 (n-3)	0.03 ± 0.02	0.03 ± 0.01	0.03 ± 0.00	0.003	0.715
C21:0	0.09 ± 0.14	0.05 ± 0.05	0.02 ± 0.01	0.020	0.359
C22:6 (n-3)	0.04 ± 0.02	0.05 ± 0.01	0.06 ± 0.01	0.004	0.496
SFA	46.39 ± 1.87	44.82 ± 1.94	44.56 ± 3.87	0.641	0.470
UFA	53.61 ± 1.87	55.18 ± 1.94	55.44 ± 3.87	0.641	0.470
PUFA	18.03 ± 3.67	17.94 ± 5.80	21.47 ± 2.17	1.160	0.276
MUFA	35.58 ± 3.89	37.24 ± 4.71	33.97 ± 3.23	0.940	0.389
n-6 PUFA	14.99 ± 2.91	13.97 ± 4.48	16.04 ± 2.31	0.791	0.576
n-3 PUFA	3.04 ± 0.83 ^B^	3.97 ± 1.49 ^ABd^	5.43 ± 0.77 ^Ac^	0.700	0.006
n-6/n-3 PUFA	5.06 ± 0.68 ^A^	3.68 ± 0.63 ^B^	3.02 ± 0.68 ^B^	0.602	<0.001

Results were presented by mean ± standard deviation (SD). SEM = standard error of the mean. (*n* = 6, number of replicates); ^c,d^ Different superscripts within a row indicate significant differences (*p* < 0.05). ^A,B^ Different superscripts within a row indicate significant differences (*p* < 0.01). ‘SFA’: saturated fatty acid, SFA = ∑ (C12:0, C14:0, C15:0, C16:0, C17:0, C18:0, C21:0); ‘UFA’: unsaturated fatty acid, UFA = ∑ (C16:1, C18:1 (n-9), C18:2 (n-6), C18:3 (n-6), C18:3 (n-3), C20:2 (n-6), C20:3 (n-6), C20:3 (n-3), C20:4 (n-6), C20:5 (n-3) C22:6 (n-3)); ‘MUFA’: monounsaturated fatty acid, MUFA = ∑ (C14:1, C16:1, C18:1 (n-9)); ‘PUFA’: polyunsaturated fatty acid, PUFA = ∑ (C18:2 (n-6), C18:3 (n-6), C18:3 (n-3), C20:2 (n-6), C20:3 (n-6), C20:3 (n-3), C20:4 (n-6), C20:5 (n-3), C22:6 (n-3)). n-6 PUFA = ∑ (C18:2 (n-6), C18:3 (n-6), C20:2 (n-6), C20:3 (n-6), C20:4 (n-6)). n-3 PUFA = ∑ (C18:3 (n-3), C20:3 (n-3), C20:5 (n-3), C22:6 (n-3)). ^1^ C16:1 = ∑ (C16:1(n-7), C16:1(n-9)). ^2^ ‘8:1, 5:1, 3:1’: different n-6/n-3 PUFA ratio diets.

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
