# Peer review of "Low Dietary n-6/n-3 PUFA Ratio Regulates Meat Quality, Reduces Triglyceride Content, and Improves Fatty Acid Composition of Meat in Heigai Pigs"

_animals, 2020, doi:10.3390/ani10091543_

Round 1

Reviewer 1 Report

The manuscript of Nong et al. reported a dietary alterations of n-6/n-3 PUFA in local pig breed (Heigai pigs), which could improve the pork quality. The topic of manuscript fits this journal. The finding of this manuscript could be beneficial to swine research as well as industry. However, revision is suggested to solidify the manuscript.

Major comments,

The authors did not measure the actual PUFA concentrations in the diet, which leads to unreliable ratio of n-6/n-3 PUFA when interpreting the treatment effects, or when comparing results with other peer researches.

The hypothesis needs to be more specific, such as the rationale of choosing such n-6/n-3 ratios as treatments, the age of pigs and experiment duration.

More details of experimental design need to be addressed, such as analysis of gene expression data.

Minor comments,

Line 7-27. Condense Author affiliation info, only contact info of corresponding author is needed.

Line 60-61. Add reference to this statement.

Line 59-90. Add references indicating the ideal animal’s age and the duration of dietary manipulation of n6/n3 PUFAs. Why did the author focus on the finishing pigs?

Line 93-101. The pigs were individually housed or group housed? Did authors use individual animal or pen as experimental unit for growth performance data analysis?

Table 1. What are the total DE of each treatment diet? What are the analyzed n-6/n-3 PUFA in each treatment diet?

Line 69-172. Why do the authors not analyze the linear (or quadratic) response of a dose treatment?

Line 153-168. Which treatment is the control diet that the authors used to calculate relative gene expression?

Reviewer 2 Report

Line 29 to 32 – is long sentence – split into two –

Line 34 pigs were fed diets with different n-3/n-6 ratios

Line 35 don’t need our results showed – Low dietary n-6/n-3 fatty acid ( don’t need to say polyunsaturated repeated times) – did not influence growth performance and carcass quality but regulated meat color, reduces … etc

Line 39 and 90 improve for better

Line 45 don’t need the results showed – just Pigs fed

Line 66 dietary fatty acid composition?

Line 78 compared with

Table 1 lone 109 – literal stone of rock powder or powdered dicalcium phosphate or limestone ? powdered mineral ?

Line 101 12 h blood sampling ?

Grass powder – dehydrated alfalfa or dried ground hay -- what is lysine content ?

Line 114 what samples – muscle and fat tissue or blood ?

Line 117 remove the tissue block – is not clear – A block of (fat or muscle tissue ) was removed and a small block of tissue was put into fixative solution –

Line 123 at the midline line 123 to 124 sentence is not complete – Backfat thickness measurements were taken at the midline. Three measurements taken at the first rib, last rib and last lumbar vertebrae were averaged.

Line 126 muscle – pH of what muscle? Line 130 to 131 – what reference card ( it is a reference card from US – Canada – Japan ) – that is public -- ? if so please reference.

Line 130 to 131 – how many values on the scores 1 to 5 ? etc

line 154 any lipid extraction on fat tissue before doing rna extraction?

table two – what is called feed conversion ratio % is really Gain: feed or feed efficiency –

table 3 – is lean percentage predicted from backfat and LMA – if predict equation is used that give a reference – “lean percentage” has a difference meaning – dissected lean – fat standardized lean, fat-tissue free lean etc –

not enough pigs to detect small to medium sized differences – between the dietary treatments.

Would have been good to have calculated Iodine value of AOAC –

Did you get estimate of the amount of lipid in the LDM ?  

Lie 235 avoid “showed” – pigs fed --- had –

Line 234 pigs fed the diet with the 5:1 ratio had the lowest n-6/n-3 ratios

Line 236 the remainder of the fatty acid concentrations

Line 256 – was expressed at greater level in the pigs fed the dietary

Line 263 avoid “in the rest” -- No significant differences in gene expression were fond for the remaining genes…

Avoid showed – don’t need “the results of our study showed “ IN this study the growth

Line 287 – in this study pig fed the dietary n-6/n-3 PUFA ratio of 3;1 had reduced

Line 292 the current and previous

Line 295 we found no significant differences for sensory evaluation of meat color.

Line 302 – avoid in line with – agree with or are in agreement with or support

Line 318 Fatty acids are released from intracellular fat stores by the action (or signaling ) or ATGL –

Our results indicate that increased dietary linseed oil percentage may reduce fatty acid release by reducing --

Line 334 this study demonstrates --

Reviewer 3 Report

Broad comments:

- The authors describe in Table 1 the composition and nutrient levels of the diet. What do the authors mean by “Nutrient levels”?

More importantly, the fatty acid profile of the diets has not been described. It can be inferred that the different ratios of n/6/n-3 were obtained changing the % of soybean oil and linseed oil. Nevertheless, it would be necessary that the authors explain how the ratios (8:1, 5:1 and 3:1) were calculated and which fatty acids were considered for such calculations. Therefore, the fatty acid profile of the diets should be included as well.

- Related to Experimental Design. Further information or clarification is needed. Although the explanation needs to be improved, it seems that a total of 54 animals were used in the experiment: 3 treatments and 18 animals per treatment. Nevertheless, the authors selected 2 animals per pen which can be very controversial attending to the statistical analysis. Most authors consider that, in the case of productive parameters (Tables 2 and 3), the real number of animals must be 9 (2 animals from the same pen should be considered as 1 because the statistical unit would be the pen). In the case of meat quality parameters (Tables 4-6 and Figures), although controversial, the animal could be considered as the statistical unit, so n = 18.

- I wonder if a different test for mean comparison was tested by the authors. Tukey test is quite strict or exigent. Perhaps more statistical differences could be found and discussed if Duncan or Bonferroni tests were considered.

I would like to see the RMSE and P-values of the statistical analyses. Why were they not shown in the tables?

In fact, the comment of L. 175-177. supports such an assumption. Such “trends” are not statistically justified in Table 2. Besides, P-values must be included in the text.

Table 2: why F:G  and Feed conversion ratio (%) have not been expressed as the other variables?
Table 3 and L. 183-185. As previously explained, from the values of SEM shown in the table, it could be inferred that statistical differences would be observed if another test was used. Therefore, the results description should be improved.

- Table 4 and L.190-195. The authors must review the statistical analysis and the description of the results. According to the values shown in Table 4, statistically significant differences should have been described for pH (24h) and Drip loss (%). Besides, the time effect (differences between 24 h and 45 min) must be also analyzed and described.

- Figure 1. Please, increase the quality of the Figure. The letters are blurry and cannot be properly read.

Besides, how did the authors analysed the images of Figure 1a? how were the differences in morphological structure analysed and stated?

- Table 5. The authors must review the statistical analysis and the description of the results. According to the values shown in Table 5, statistically significant differences should have been described for C14:0, C16:1, C18:0, C10:3n-6, C22:6n-3, n-6 PUFAs.

Therefore, the description of the results (L.226-237) should be improved.
- Table 6 and L. 244-253: Please see above. Same comments as table 5 are applied.

- Figure 2 and L. 255-263. Please, review the data and analyses of Figure 2a LDM, Figure 2b SAT, and Figure 2d LDM. No statistically significant difference was detected because a big value of SEM (high variability) was shown. Do the authors have any explanation for such behaviour?

Besides, please, review Figure 2c LDM stats because if seems pretty obvious that 5:1 resulted different than the others.

Accordingly, please, improve the description of the results (L. 244-253).

Please, the Results section must be improved. More effects and differences will be presented if a better look at the data was carried out.

- Discussion section. This section must be improved and rewritten. It seems a mere description of the results (again) followed by a list of different papers that agree or disagree with the results.

Please, improve this section by describing the motive of disagreement or agreement.

Please, also consider that the considered pigs were slaughtered at a lower weight, lower fat thickness, etc… than most of the considered bibliography.

- 41 references have been used. It is a quite number. Some of them are useless and redundant.

For example, in the introduction 16 references were used but a third of them are redundant.

Another example, in 2.7 Fatty acid composition analysis, 3 references were used and they are redundant. Nevertheless, no reference was included in 2.8 Total RNA extraction, reverse transcription and quantitative real-time PCR.

Specific comments:

  • 44. 3 replications or 3 replicas? The whole sentence must be clarified (see comment above).
  • Table 1: Digestive energy values lacks of SEM.
  • Table 1 and table footnote (L.105-111): Superindexes must be reviewed.
  • 111, 181. Please, include another superindex for such a comment.
  • Table 1: Bran? Meaning? Wheat bran? Bran cereal? Rye bran? Oat bran?
  • Table 1: The “Total” row can be deleted.
  • 113. This sentence or statement makes no sense. Please, review.
  • 128-130. 24 h? The cited methodology describes more than 24h… ¿?
  • 131. cards?
  • 222-224. Please, include the meaning of SFA, UFA, MUFA, PUFA.
    The row of “total” in table 5 can be deleted (same for Table 6).
  • Tables 5 and 6. n-9 instead of n9 for C18:1, C22:1.

Besides, in Table 5, C20:1 must be C20:1n-9. C16:1 must be the sum of n-7 + n-9 (clarify!), C20:2 n-????

Attending to Table 6, C14:1 must be C14:1n-5. C16:1 must be the sum of n-7 + n-9 (clarify!); C20:2 n-????; C20:0 and C20:1n-9 were not shown but C20:2 was?

Round 2

Reviewer 1 Report

The authors did not address review comments accordingly in the revised version. Major comments,
1. The accurate composition of PUFA in the diet is critical to the present study. However, the authors do not have a clear result indicating the total PUFA, PUFA profile, and n6/n3 ratio between treatments. Unclear PUFA composition in diet can lead to confusions in interpreting other results, as well as in comparison with other peer studies.
2. If the gene expression data were normalized to 8:1 diet group, then the relative mRNA of the control (8:1 group) should be very close to 1. However, it is not the case in figure2. Please explain.
3. Author affiliation 1-6 and 7-8 should be combined. Contact info (email address) should be only listed for corresponding authors

Author Response

Response to the comments

--Thanks for your valuable comments! We are grateful to you for the critical comments that helped to improve our manuscript. The manuscript has been revised according to the comments. Please find our point-to-point response to the comments.

Response to Reviewer:

The authors did not address review comments accordingly in the revised version. Major comments,

-- Thank you for your suggestion. We will do our best to modify as required.

  1. The accurate composition of PUFA in the diet is critical to the present study. However, the authors do not have a clear result indicating the total PUFA, PUFA profile, and n6/n3 ratio between treatments. Unclear PUFA composition in diet can lead to confusions in interpreting other results, as well as in comparison with other peer studies.

-- Fatty acid profiles of diets have been added in the revised manuscript. (Table A1)

  1. If the gene expression data were normalized to 8:1 diet group, then the relative mRNA of the control (8:1 group) should be very close to 1. However, it is not the case in figure2. Please explain.

-- Thanks! Individual differences (intragroup variability) may have some influence in the results, but these are the true results of our experiments.

  1. Author affiliation 1-6 and 7-8 should be combined. Contact info (email address) should be only listed for corresponding authors.

-- Author affiliation has been combined and contact info (email address) of other author has been deleted in the revised manuscript. (line 7-11)

Reviewer 2 Report

still some grammar issues - verb tenses 

Author Response

Thank you for your valuable comments! We are grateful to you for the critical comments that helped to improve our manuscript. The manuscript has been revised and improved. Please see the revised manuscript. Thank you again and wish you a happy life.

Reviewer 3 Report

The authors have taken into account all the comments. The quality of the manuscript has increased exponentially.

Nevertheless, some other issues have to be reviewed before publication:

  • I still have strong doubts or considerations related to Tables 2, 3 and 4 statistics. But, because the authors explained that a review and confirmation of SEM, P and n values were carried out, the only explanation for my considerations should be related to a different SD or variability for each group.
    Could the authors study the intragroup variability?
    In fact, a different variability among treatments would be a differentiating result.
    Better if a relationship between dietary PUFA ratios and enzyme expression with the different variabilities was found.
  • Table 4. Why weren’t the time differences of colour variables considered?
    Please, review the Results and the Discussion sections if needed.
  • The authors said that the quality of Figures 1 and 2 was increased but it seems that just a size increase was considered.

In Figure 1 a: it is impossible to read the “scale”. Besides, some red squares can be observed. Meaning?

Figure 1 b, c and d. Several squares, triangles and circles are shown for each diet. Meaning? Why wasn’t the same nomenclature of Figure 2 considered in Figure 1? The squares, triangles and circles are useless. In fact, they hinder the adequate observation of the SEM lines. Please, remove all the black squares, triangles and circles and use the same key of Figure 2.

  • 171. The sentence is not correct. Please, correct it. As a suggestion: “The lower the dietary n-6/n-3 PUFA ratio considered, the higher the value of F:G ratio was observed. Besides, a difference of 2.41% was observed between 5:1 and 3:1 ratios but 8.92% between 8:1 and 5:1.” (Table 2).

Could the authors provide an explanation for such a different behaviour?

  • Please, confirm with the editorial office that the use of “Longissimus dorsi muscle” is accepted. Many journals are no longer accepting such a name and requiring the use of “Longissimus Lumborum and Thoracis muscle”. Besides, confirm whether or not it has to be in italics.
  • There is no need to repeat the meaning of each colour variable every time. “Lightness L*” can be substituted by L*, “redness a*” by a* and “yellowness b*” by b*.

Minor comments:

  • Please, include the SD in “content” values from Table 1.
  • It should be considered to remove the “s” from the abbreviation “PUFAs”.
  • 20-21. The sentence “To investigate the effects of dietary supplementation with different n-6/n-3 PUFAs ratios on growth performance, meat quality and fatty acid profile in Heigai pigs.” is grammatically wrong (lacking verb and predicate) and makes no sense. Either consider to delete it or to rephrase L.20-22.
  • 24. Of pork? Sounds weird.
  • 28. Fatty acidS ratio is not correct.
  • 52, 76, 104... (Dietary) fatty acidS composition is not correct. Please, correct the recurrent mistake of L.28 and L.52.
  • 101. Meat quality parameters.
  • 108. The abbreviation “BW” must be previously described (L. 80 perhaps?).
  • 117. When was “marbling” measured?
  • 137. ContentS?

Author Response

Response to the comments

The authors have taken into account all the comments. The quality of the manuscript has increased exponentially. Nevertheless, some other issues have to be reviewed before publication:

-- Thank you for your valuable comments! We are grateful to you for the critical comments that helped to improve our manuscript. The manuscript has been revised according to the comments again. Please find our point-to-point response to the comments.

Response to Reviewer:

  1. I still have strong doubts or considerations related to Tables 2, 3 and 4 statistics. But, because the authors explained that a review and confirmation of SEM, P and n values were carried out, the only explanation for my considerations should be related to a different SD or variability for each group.

Could the authors study the intragroup variability?

In fact, a different variability among treatments would be a differentiating result.

Better if a relationship between dietary PUFA ratios and enzyme expression with the different variabilities was found.

-- Thanks! “SD” has been added in all tables. The relationship between dietary PUFA ratios and enzyme expression with the different variabilities will be considered in the next research work (ongoing).

  1. Table 4. Why weren’t the time differences of colour variables considered?

Please, review the Results and the Discussion sections if needed.

-- Time differences of color variables result and discussion have been added in the revised manuscript. (Table 4 and line 296-298)

  1. The authors said that the quality of Figures 1 and 2 was increased but it seems that just a size increase was considered.

In Figure 1 a: it is impossible to read the “scale”. Besides, some red squares can be observed. Meaning?

-- The “scale” has been moved to figure legend (line 202). “Red squares” is fat cell membrane. The quality of Figures 1 and 2 was increased in the revised manuscript (Figures have been revised and pdf documents (maybe clearer) have been sent to the editor).

  1. Figure 1 b, c and d. Several squares, triangles and circles are shown for each diet. Meaning? Why wasn’t the same nomenclature of Figure 2 considered in Figure 1? The squares, triangles and circles are useless. In fact, they hinder the adequate observation of the SEM lines. Please, remove all the black squares, triangles and circles and use the same key of Figure 2.

-- The black squares, triangles and circles have been removed and the key has been changed to the same key of Figure 2.

  1. The sentence is not correct. Please, correct it. As a suggestion: “The lower the dietary n-6/n-3 PUFA ratio considered, the higher the value of F:G ratio was observed. Besides, a difference of 2.41% was observed between 5:1 and 3:1 ratios but 8.92% between 8:1 and 5:1.” (Table 2).

Could the authors provide an explanation for such a different behaviour?

--The sentence has been edited according to the comments. The explanations for the different behavior of F:G ratio: â‘ They are not completely arithmetic series (8:1 to 5:1, 5:1 to 3:1); â‘¡F: G ratio and diet n-6/n-3 PUFA ratio are not completely negatively correlated (There may exist a certain proportion has greater influence); â‘¢May be affected by other factors.

  1. Please, confirm with the editorial office that the use of “Longissimus dorsi muscle” is accepted. Many journals are no longer accepting such a name and requiring the use of “Longissimus Lumborum and Thoracis muscle”. Besides, confirm whether or not it has to be in italics.

--Thank you for your suggestion! We have sent an email to the editor for confirmation. “Longissimus dorsi muscle” is accepted (normal or italic).

  1. There is no need to repeat the meaning of each colour variable every time. “Lightness L*” can be substituted by L*, “redness a*” by a* and “yellowness b*” by b*.

-- Redundant words have been deleted in the revised manuscript.

Minor comments:

  1. Please, include the SD in “content” values from Table 1.

-- “SD” values have been included in Table 1.

  1. It should be considered to remove the “s” from the abbreviation “PUFAs”.

-- All “s” has been removed from the abbreviation “PUFAs”.

  1. 20-21. The sentence “To investigate the effects of dietary supplementation with different n-6/n-3 PUFAs ratios on growth performance, meat quality and fatty acid profile in Heigai pigs.” is grammatically wrong (lacking verb and predicate) and makes no sense. Either consider to delete it or to rephrase L.20-22.

--Thanks for your suggestion! The sentence has been rephrased in the revised manuscript. (line 16-19)

  1. Of pork? Sounds weird.

-- “Of pork” has been changed to “of LDM and subcutaneous adipose tissue”. (line 21)

  1. Fatty acidS ratio is not correct.

--“Fatty acidS ratio” has been replaced by “fatty acid ratios”. (line 25)

  1. 52, 76, 104... (Dietary) fatty acidS composition is not correct. Please, correct the recurrent mistake of L.28 and L.52.

--All “fatty acidS composition” have been change to “fatty acid composition”. (line 49, 58, 72, 99…)

  1. Meat quality parameters.

-- According to the context, we found the sentence should be deleted, so the whole sentence has been deleted.

  1. The abbreviation “BW” must be previously described (L. 80 perhaps?).

--Thanks! “BW” has been previously described. “BW” in line 108 has been replaced by “Slaughter weight” (used in Table 3). (line 81 and 103)

  1. When was “marbling” measured?

--The “marbling” measured at 45 min after slaughter and the details have been added in the revised manuscript. (line 113)

  1. ContentS?

--“Content” has been replaced by “contents” in the revised manuscript. (line 133)

Round 3

Reviewer 3 Report

  • L. 7. There is no need to use such amount of numbers: 1, 2,3,4,5,8 could be substituted by just "1".
  • L. 10. As above, 6 and 7 could be substituted by "2".

Please, review L. 5 and 6 accordingly.